# Neutrophils in Cancer and Potential Therapeutic Strategies Using Neutrophil-Derived Exosomes

**DOI:** 10.3390/vaccines11061028

**Published:** 2023-05-26

**Authors:** Abhishek Dutta, Shrikrishna Bhagat, Swastika Paul, Jonathan P. Katz, Debomita Sengupta, Dharmendra Bhargava

**Affiliations:** 1Exsure, Kalinga Institute of Industrial Technology, KIIT Rd, Patia, Bhubaneswar 751024, Odisha, India; 2Department of Gastroenterology, 928 BRB II/III, 421 Curie Blvd, University of Pennsylvania, Philadelphia, PA 19104, USA; 3Department of Environmental Carcinogenesis & Toxicology, Chittaranjan National Cancer Institute (under Ministry of Health and Family Welfare, Government of India Regional Cancer Centre), 37, S.P. Mukherjee Road, Kolkata 700026, West Bengal, India

**Keywords:** Neutrophils, tumor-associated neutrophils, neutrophil-derived exosomes, immune suppression, metastasis, tumor microenvironment, neutrophil-extracellular trap, trogocytosis, ADCC and ADCP

## Abstract

Neutrophils are the most abundant immune cells and make up about 70% of white blood cells in human blood and play a critical role as the first line of defense in the innate immune response. They also help regulate the inflammatory environment to promote tissue repair. However, in cancer, neutrophils can be manipulated by tumors to either promote or hinder tumor growth depending on the cytokine pool. Studies have shown that tumor-bearing mice have increased levels of neutrophils in peripheral circulation and that neutrophil-derived exosomes can deliver various cargos, including lncRNA and miRNA, which contribute to tumor growth and degradation of extracellular matrix. Exosomes derived from immune cells generally possess anti-tumor activities and induce tumor-cell apoptosis by delivering cytotoxic proteins, ROS generation, H_2_O_2_ or activation of Fas-mediated apoptosis in target cells. Engineered exosome-like nanovesicles have been developed to deliver chemotherapeutic drugs precisely to tumor cells. However, tumor-derived exosomes can aggravate cancer-associated thrombosis through the formation of neutrophil extracellular traps. Despite the advancements in neutrophil-related research, a detailed understanding of tumor-neutrophil crosstalk is still lacking and remains a major barrier in developing neutrophil-based or targeted therapy. This review will focus on the communication pathways between tumors and neutrophils, and the role of neutrophil-derived exosomes (NDEs) in tumor growth. Additionally, potential strategies to manipulate NDEs for therapeutic purposes will be discussed.

## 1. Introduction

Neutrophils are the most common polymorphonuclear (PMN) leukocytes [1] which constitute around 40% to 70% of total white blood cells. Neutrophils act as an immediate innate immune responder against inflammation [2], injury and infection [3]. Invading pathogens invoke an inflammatory response that recruits neutrophils to the site of inflammation following chemical signals like IL-8, Leukotriene B4, C5a or H_2_O_2_ and destroys the pathogen by phagocytosis and release of antimicrobial substances like neutrophil extracellular traps (NETs) [1]. In response to pathogens or inflammatory molecules, neutrophils get activated, marginalize themselves onto the walls of blood vessels, and invade infected or inflamed tissue, frequently even in tissue environments where other molecules or cells are unable to access and begin to release proteinases, cytokines and chemokines which direct the neighboring cells function towards resolution of inflammation [4] as well as towards initiating an immune response to clear off the pathogen [5]. Previously, neutrophils were considered pathogen-clearing cells. However, over the last two decades, neutrophils have been recognized as a highly versatile and sophisticated population, displaying both heterogeneity and plasticity. Based on the pathological niche, neutrophils can differentiate into distinct subsets like immune-suppressive neutrophils in HIV and cancer or pro-inflammatory neutrophils in SLE or psoriasis [6]. The niche dependent differentiation of neutrophils and their elevated count in cancer attracted the attention of cancer biologists in order to uncover the role of neutrophils in cancer. In most cases, immune cells can be identified by their specific surface markers. Neutrophils lack any known well-defined specific surface marker which can distinguish neutrophils from other immune cells. A number of suggested markers, including LDL receptor 1 [7] and CD10 [8], are under investigation. However, poorly defined surface markers, difficulties during isolation and short half-life are the reasons for the difficulties of neutrophil study. 

Tumor microenvironment (TME) is characterized by chronic inflammation which is considered one of the hallmarks of cancer. Approximately 25% of tumor cases are characterized by chronic inflammation and infections caused by tumors [9]. Tumor inflammation is largely promoted by infiltrating immune cells. Along with other immune cells, neutrophils are a major inflammatory cell in solid tumors [10] such as melanoma, head and neck squamous cell carcinomas (HNSCCs), bronchoalveolar carcinomas, and renal carcinomas [11]. Elevated numbers of neutrophils in the blood [12] and tumors [13] are associated with a poor prognosis. 

Tumor-associated neutrophils (TANs) seem to be an effective way to treat cancer, given the association between TAN infiltration and high neutrophil-to-lymphocyte ratio (NLR) and a poor prognosis and low recurrence-free survival. Notably, NLR correlation with prognosis is a misleading term because neutrophils are double-edged swords and their phenotype strictly depends on the location of neutrophils, tumor niche, stage and type of tumor. There are, however, several drugs that target neutrophils already approved by the FDA to treat various inflammatory and autoimmune diseases, such as vasculitis, gout, rheumatoid arthritis, and cancer [14,15]. Some examples of anti-neutrophil drugs are acetylsalicylic acid (aspirin), which inhibits prostaglandin synthesis by neutrophils, and CCX168, which blocks the complement 5a receptor (C5aR) on neutrophils and reduces their activation and recruitment [16]. However, one of the challenges in anti-neutrophil drug design is the difficulty in identifying and targeting different subtypes of tumor-associated neutrophils (TANs), such as the pro-tumoral N2 TANs, which have distinct phenotypes and functions from the anti-tumoral N1 TANs. Therefore, more specific and selective anti-neutrophil drugs are needed to effectively modulate neutrophil function in the tumor microenvironment.

Exosomes are nano-sized vesicles that are released by neutrophils and other cells and can modulate intercellular communication by transferring proteins, lipids, and nucleic acids. In asthma, neutrophil-derived exosomes (NDEs) can enhance the proliferation of airway smooth muscle (ASM) cells, which contributes to airway remodeling and obstruction. NDEs can also affect other immune cells in the airway, such as macrophages, dendritic cells, and T-cells, by altering their cytokine production, antigen presentation, and activation status demonstrating a new mechanism that links neutrophils to immune responses and tissue remodeling in asthma. In recent years, exosomes have become a hot topic among cancer biologists for modulating the TME. Exosomes, in general, contain various immunomodulatory molecules like TGFβ, IL10, Foxp3, IL6, and PGE2, which have the potential to disarm active immune cells like CD4^+^ T-cells, NK cells, and macrophages. On the other hand, tumor cell-derived exosomes (TE) are capable of recruiting immune suppressive T-regulatory cells in the tumor microenvironment [17]. TEs can also drive neutrophil polarization towards the N2 subtype and induce autophagy via HMGB1/TLR4/NF-κB signaling pathway [18]. Tumor stem-like cells (TSLC)-derived exosomes contain various RNAs, such as microRNAs, long non-coding RNAs, and circular RNAs. These RNAs can modulate their gene expression and function. In colorectal cancer, TSLC-derived exosomal RNAs can prime neutrophils to acquire a pro-tumoral N2 phenotype [19], which is characterized by increased expression of arginase-1, vascular endothelial growth factor, and matrix metalloproteinase-9. However, whether these TEs driving neutrophil polarization are amenable to target, is subject to further studies exploring tumor-neutrophil signaling networks.

In addition to being subject to targeted therapy, exosomes have emerged as an efficient drug-delivery system owing to their desirable intrinsic features, such as biocompatibility, nano-scale effects, long-range targeting, and stability in circulation. Today exosomes are being engineered to deliver anti-tumor agents directly into tumor cells to reduce drug-mediated toxicity. Neutrophil-derived exosomes (NDEs) are known to induce anti-tumor effects via activation of the apoptotic signaling pathway [20]. NDEs can efficiently cross the blood–brain barrier (BBB) proving to be an effective therapeutic approach to target inflamed brain tumors [14]. Dendritic cell-derived exosomes (DCE) have been shown to promote tumor cell apoptosis through TNF superfamily ligands [21]. Natural killer cell-derived exosomes (NKE) have been reported to exert cytotoxic effects through FasL and perforins [22]. Neutrophil-derived exosomes have been shown to act against arthritis and sepsis setup. However, except for being used as an effective way of targeting tumors in mice [20], the potential of this method as a therapeutic tool has not been extensively investigated due to its limited lifespan and low productivity. In this review, we aim to highlight the role of neutrophils in manipulating tumor-neutrophil crosstalk, as well as possible ways to target tumor cells and associated neutrophils using exosomes.

## 2. Neutrophils: From Generation to Tumor Infiltration

Neutrophils are generated in the bone marrow every day from hematopoietic stem cells [1] through lymphoid priming under the stimulation of granulocyte colony-stimulating factor (G-CSF), which, in a recombinant version, is used to combat neutropenia developed after chemotherapy. The maturation process (Figure 1) includes stages of promyelocyte, myelocyte, metamyelocyte band cell and finally neutrophil with the nucleus transforming from a round shape to a lobed morphology [23] and changing expression markers. In neutrophil differentiation, transcription factors play a crucial role, and their expression varies with the stages of differentiation. *PU.1* and *C/EBPα* are essential for myeloid commitment and expression of the G-CSF receptor. *C/EBPδ* and growth factor independent-1 (*Gfi-1*) promote terminal differentiation [24,25]. *C/EBPβ* controls emergency granulopoiesis [26]. *Runx1^-/-^* and *Klf6^-/-^* neutrophils cannot migrate and recruit at an inflamed site. *RELB, IRF5* and *JUNB* drive neutrophil effector responses. *RFX2* and *RELB* are required for neutrophil survival during inflammation [27]. The various stages of differentiation can be also identified by the existence of different types of granules like azurophil in myeloblast to promyelocyte or gelatinase granules during the band cell stage. The mentioned granules contain myeloperoxidase (MPO) and elastase, proteolytic enzymes like cathepsin-G, proteinase-3, antimicrobial defensins and matrix metalloproteinases. Bone marrow stromal cells produce ligands specific to the receptors displayed by neutrophil progenitors for the same to be retained in place. These ligands include VCAM-1 against VLA4 and CXCL-12 against CXCR4 [28]. Mature neutrophils in the bone marrow are kept in place through the action of the chemokine receptors CXCR2 and CXCR4 [29]. G-CSF induces the exit of mature neutrophils from bone marrow through the disruption of CXCL12 and CXCR4 interaction, while ligands of CXCR2 are demonstrated by the endothelial cells of blood vessels. G-CSF also reduces CXCL12 expression in bone marrow stromal cells as well as CXCR4 in neutrophil. IL-17 is responsible for the production of G-CSF which involves positive feedback loops along with IL-1 and IL-23. Neutrophils are thereby mobilized to the site of inflammation through the leucocyte adhesion cascade. Noteworthy, studies in murine models and patients regarding neutrophil recruitment differ in chemokine receptor expressions in mice vs. humans, and, therefore, lack of response observed in Lewis lung cancers implanted into CXCR2^-/-^ or CXCR2^+/+^ mice may result from different pathways.

The endothelial cells of the blood vessel at or near the inflamed site express E and P-selectins which bind to the glycoprotein ligands on neutrophil making them roll on the endothelium. Eventually, ICAM-1 and ICAM-2 ligands of the endothelial cells bind to the β2 integrin receptors on the neutrophils facilitating their adherence to the wall of the blood vessel. 

The mechanisms of tumor-induced neutrophilia are unclear, but tumor-derived cytokines such as G-CSF, IL-1 and IL-6 may stimulate granulopoiesis or neutrophil development. Two populations of neutrophils are found outside bone marrow: the circulating ones and the marginalized ones [30]. The circulating ones are induced from the marginalized ones and recruited to the site of inflammation or the tumor. Activated neutrophils in the peripheral tissue are expected to follow chemoattractants and show N1 phenotype in the tumor context. Chemokines such as IL-8, CXCL6, and CCL3 attract neutrophils to tumor sites [31]. Ras activates CXC chemokines, which increase neutrophil infiltration in mice [31]. CXCL5 (epithelial neutrophil-activating peptide-78) and MMP-9 also recruit neutrophils in Hepato-Cellular Carcinoma (HCC) and promote tumor progression and metastasis [31]. In breast cancer mice models, IL-17 has been shown to bring neutrophils into the TME [32]. IL-17 also has roles in antitumor responses and N2 polarization, which will be discussed later.

Human tumors contain TANs with different phenotypes: N1 (CD62L^low^ CD54^hi^) and N2 (CD45^+^ Lin^−^ HLADR^−^ CD11b^+^ CD33^+^ CD66b^+^ [33], which are similar to G-MDSCs. The N2 phenotype may arise from the N1 phenotype or vice versa, depending on the levels of TGFβ, G-CSF or type-I Interferons in the TME. TGFβ promotes the conversion of immature and immunosuppressive LDNs to N2 phenotype, which resembles G-MDSCs. Tumor-induced neutrophilia also correlates with increased immature myeloid cells.

## 3. Tumor-Associated Neutrophils

Tumor-associated neutrophils (TANs) can be subdivided into two subtypes like tumor-associated macrophages (TAMs)-M1 and M2; namely anti-tumorigenic N1 (Good) and pro-tumorigenic N2 (Bad) (Figure 2). Good TANs act as anti-tumorigenic when they acquire the N1 phenotype, which is characterized by killing tumor cells directly by releasing granule contents such as elastase, cathepsin G and myeloperoxidase (MPO), or indirectly by forming neutrophil extracellular traps (NETs) that trap and damage tumor cells and secretion of immunostimulatory cytokines and chemokines such as interferon-γ (IFN-γ), tumor necrosis factor-α (TNF-α) and IL-12, or by presenting tumor antigens to T-cells [34]. Factors that can induce the N1 phenotype are interferon-β (IFN-β), tumor necrosis factor-α (TNF-α) and transforming growth factor-β (TGF-β) inhibitors [31,35]. These cytokines can induce the expression of pro-inflammatory molecules and cytotoxic mediators on neutrophils, such as ICAM-1, IP-10/CXCL10, TNF-α, elastase, and cathepsin G. N1 neutrophils can be generated by exposure to tumor cell lysates or apoptotic tumor cells, which can stimulate the production of interferon-γ (IFN-γ) and tumor necrosis factor-α (TNF-α) by natural killer (NK) cells and T-cells [36,37]. Therefore, tumor cell lysates or apoptotic tumor cells can induce a positive feedback loop between N1 neutrophils and other immune cells to inhibit tumor progression. Some other anti-tumorigenic effects of neutrophils include an inhibiting angiogenesis, invasion and migration of tumor cells by expressing antiangiogenic factors such as angiostatin, endostatin and thrombospondin-1 [34], the prevention of metastatic colonization by blocking adhesion molecules on endothelial cells or by competing with tumor cells for attachment sites [34].

Bad neutrophils (N2 TANs) are substantially different from the N1 subtype in their activation and cytokine status. In addition, the N2 population has different morphology from the N1 subtype, with its hypersegmented nuclei compared to the circular nuclei of N1 neutrophils [38]. N2 TANs promote tumor development and progression. However, the exact mechanisms of the actions of N2 TANs are not fully understood. It is believed that N2 TANs stimulate immunosuppression, tumor growth, angiogenesis, and metastasis by DNA instability, or by cytokines and chemokines release [31,39]. N2 TANs mainly secrete immunosuppressive cytokines, matrix-degrading proteases and angiogenesis-promoting molecules [40] and thus cause immune escape. Overall, the presence of N2 TANs in the tumor microenvironment is associated with poor prognosis in various types of cancer, including lung cancer, breast cancer, and colorectal cancer. However, more research is needed to fully understand the complex interactions between N2 TANs and other immune and tumor cells, as well as to develop effective strategies to target them for cancer therapy. In addition to serving as pathogenic markers, the neutrophil to lymphocyte ratio (NLR) has been already declared to be an immensely important feature to predict fatal complications [41].

Some studies suggest that N1 and N2 TANs are plastic and can be polarized by different stimuli [31,38]. N1 polarization can be induced with LPS IFN-β, IFN-γ [42,43,44] and blockage of TGFβ. However, the N2 polarization requires TGFβ, IL10 PGE2, G-CSF, L-lactate and Adenosine [40,45,46,47,48,49].

Practically, the N1 and N2 phenotypes correspond to high-density neutrophils (HDNs) and low-density neutrophils (LDNs) [50]. LDNs are uncommon in non-pathological conditions but have been found to increase in metastatic disease and in patients who do not respond to neoadjuvant chemotherapy. This increase in LDNs is associated with a decrease in cytotoxic T-cell infiltration and an increase in regulatory cells. Both LDNs and HDNs express CD11b, CD66b, and CD15, with higher expression levels observed in LDNs. To differentiate between the two, mean fluorescence intensities in flow cytometric analyses need to be compared, along with using healthy donor blood as a reference. In healthy mice, approximately 95% of circulating neutrophil populations consist of HDNs [13]. However, with tumor progression, the LDN population increases while the amount of HDNs in the bone marrow remains relatively stable. This increase in LDNs is accompanied by changes in shape and activation. The characteristics of LDNs and HDNs in human and mouse populations were found to exhibit similar behavior in the presence of tumors, indicating that mouse models are suitable for analyzing these neutrophil subsets. Nevertheless, there are differences between humans and mice that need to be considered. In humans, two subtypes of neutrophils are present in the bone marrow, namely immature and mature neutrophils. In mice, an additional subtype called preNeu, which refers to proliferative neutrophil precursors, exists in the bone marrow [51]. Furthermore, there are other variations in subpopulations, such as CD177 and TCRαβ+ in human blood and the red pulp subpopulation in the mouse spleen, which should be considered during analyses. Both LDNs and HDNs were found to be derived from bone marrow and from each other, demonstrating plasticity, and suggesting future research focusing on directing tumor-induced LDNs into HDNs. The cause of mature and immature LDNs in cancer is still unclear, despite the identification of transcription factors like Gfi-1, Lef-1, C/EBPs, and Klf5, which are important for the terminal differentiation of neutrophils [52]. The mentioned factors are therefore subject to further research for risk assessment. Oxidative stress, inflammation, and systemic infection have been shown to trigger granulocytopoiesis, which compensates for the increasing demand for neutrophils. However, these factors have not yet been correlated with tumor-induced immature LDN generation. It is worth mentioning that the isolation of neutrophils along with the peripheral blood mononuclear cell (PBMC) fraction may introduce variations in terms of maturation levels, activation states, and degranulation status. Therefore, it is essential to conduct detailed phenotypic analyses and identify the tissue source from which the neutrophil population might have originated. Again, LDNs are often defined as PMN-MDSCs without a thorough evaluation of their suppressive functions on T-cells. The threshold for defining the degree of suppressive activity necessary to differentiate LDNs from normal-density neutrophils (NDNs) remains to be determined. This lack of clarity contributes to the controversies surrounding the apparent T-cell-promoting or inhibiting functions of LDNs in cancer patients. In contrast, studies conducted in mice have reported the presence of immature LDN populations with well-defined suppressive properties in blood, spleen, bone marrow, and tumor tissues.

## 4. Intra-Tumoral Activities of Neutrophils Is as Expected

The tumor microenvironment (TME) is composed of a variety of cells, including tumor cells and their surrounding immune cells, tumor-associated fibroblasts, vascular endothelial cells, etc. These immune cells include innate immune cells (neutrophils, macrophages, natural killer cells, dendritic cells, etc.) and adaptive immune cells (T and B cells). Innate immune cells contribute to tumor suppression either through direct recognition and killing of cancer cells or by triggering a strong adaptive immune response by triggering self-activation [53]. Interestingly, innate immune cells have been found to be promising tools for treating hematopoietic malignancies and solid tumors owing to their antitumor immunocompetence [54]. Neutrophils are professional phagocytes in protecting the host against microbial infection as well as inducing inflammation. Neutrophil-mediated phagocytosis of B-cell lymphoma cells has been shown to be induced by Rituximab (CD20). Antibody-dependent cellular phagocytosis (ADCP) is in fact, one of the most important methods of clearance of tumor cells. Neutrophil-mediated ADCP has been reported for a variety of therapeutic monoclonal antibodies, including Obinutuzumab, Ofatumumab, and Trastuzumab. Although preclinical studies have confirmed the role of ADCP as a potent mechanism to eliminate tumor cells, clinical studies are yet to be conducted to ensure its safe and effective use. Antibody-based activation of the complement system as well stimulates neutrophil-mediated anticancer immune responses. C3a and C5a are potent neutrophil chemoattractants that are formed during complement cascade, increasing tumor infiltration by neutrophils during antibody immunotherapy in mice models [55]. 

In principle, neutrophils cannot engulf complete cancer cells (~15 µM–25 µM in diameter) [56] which are typically larger than neutrophils (~8.85 + 0.44 µM in diameter) in humans [57]; however, once they invade tissues and attach to extracellular matrix or target cells, neutrophils can attain larger size and shape (10 µM–15 µM) [58]. An extravasating neutrophil becomes elongated due to microparticle deposition. LFA-1 (an in-tegrin, CD11a/CD18) is localized at the trailing edge of neutrophils during the elongation step of extravasation upon stimulation with fMLP and the cell length repeatedly becomes longer until the trailing edge is finally retracted in the endothelial basement membrane [59]. To subjugate this challenge neutrophils have devised a strategy called trogocytosis. In trogocytosis, a small part of the plasma membrane of the target cell is phagocytosed. This sequential trogocytosis leads to loss of membrane integrity that finally leads to lysis of the cell. In vitro demonstration of trogocytosis has been conducted for several clinical antibodies like trastuzumab, cetuximab, rituximab, etc. Trogocytosis requires physical proximity between neutrophils and tumor cells. This interaction is dependent on both CD18/CD11b and FcγR interactions. Cell death due to trogocytosis is independent of other effector mechanisms that are employed by neutrophils. However, the clinical contribution of neutrophil-mediated trogocytosis in eliminating tumor cells is yet to be determined. 

ADCC is another weapon used by neutrophils to kill cancer cells. In ADCC, neutrophils recognize the cancer cell through an antibody bound to it, and then latch onto the cancer cell and release a cocktail of chemical mediators that destroy the cancer cell. ADCC triggers the release of ROS in addition to other cytotoxic molecules such as lactoferrin, elastase, arginase, myeloperoxidase, cathepsins, defensins, and MMP9 to eliminate tumor cells. Observation has been made that the specific role played by each neutrophil Fc receptor in mediating ADCC varies depending on the type of cancer. IgG-mediated ADCC is also influenced by the local inflammatory condition. Different therapeutics are currently being used in clinics that use the ADCC pathway. Rituximab induces ADCC in B cell lymphoma [60] and cetuximab in head and neck cancer via activation of neutrophils [61]. Moreover, certain antibodies, including Trastuzumab (HER2/Neu), Zalutumumab (EGFR), and Alemtuzumab (CD52) also eliminate tumor cells by means of neutrophil-dependent ADCC. Studies have ascertained that N2 neutrophils produce and release genotoxic substances like ROS and NO into the TME which increases DNA instability [62]. An interesting observation was made by Haqqani et al., which demonstrated that the number of infiltrating neutrophils into TME correlates with the number of mutations at the hypoxanthine phosphoribosyl transferase (*Hprt*) locus in murine subcutaneous tumors. These mutations are mainly driven by ROS and iNOS released by neutrophils [63]. Myeloperoxidase produced by neutrophils is also an important mediator of genotoxicity. Hypochlorous acid produced by myeloperoxidase is one of the major drivers of mutation in the *HPRT* gene. 

Sun et al. demonstrated that neutrophils suppress tumor cell proliferation via Fas/FasL pathway-mediated cell cycle arrest. They used Fas-knockout cells to inhibit the FasL/FasR pathway in A549 cells [64,65]. Additionally, NETs also have cytotoxic effects composed of DNA-histone protease complexes, cathepsin G, and neutrophil elastase. Defensins, another component of NET, can also cause tumor cell lysis, further epithelial cells, and blood vessels that support tumor growth may be destroyed by histones in NETs [66]. Neutrophils express cytokine receptors like conventional cytokine receptors (type I and type II), TNF-receptor superfamily members, and members of the IL-1-receptors, which may assist in anticancer immune responses. TNF-α was identified as a key regulator for the cytotoxic activity of neutrophils toward breast cancer cells by means of neutrophil transmigration and NO release that promotes cancer cell killing. IL-8 and CXCL5 secreted by renal cell carcinoma (RCC) cells have been shown to recruit neutrophils and abrogate the formation of metastases supporting the anti-tumor role of neutrophils [67].

Another aspect of neutrophil intra-tumoral functions is angiogenesis. Tumor angiogenesis, primarily driven by VEGF, is facilitated by tumor-associated neutrophils (TANs) through secretion. Previous studies have reported vascularization promoted by neutrophils in various contexts, such as the proliferative stage of the endometrial cycle [68], cornea [69], and chorioallantoic membrane [70]. In the context of tumors, anti-inflammatory cortisone treatment has been shown to significantly reduce angiogenesis by 80% [71]. Furthermore, in a significant study using a pancreatic islet model of RIP1-Tag2 transgenic mice, neutrophils have been observed to specifically participate in angiogenesis [72], making them a potential target for anti-GR therapy. In this regard, the role of neutrophil-derived MMP9, which lacks the inhibitor TIMP-1 that would otherwise render MMP9 inactive as part of a complex, cannot be overlooked. TIMP-1-free MMP9 is capable of cleaving VEGF and FGF2, which are sequestered in the extracellular matrix, into active forms, thereby promoting pro-angiogenic signaling. CXCL8 and CXCL1, aside from being neutrophil chemoattractants, also function as pro-angiogenic chemokines secreted by neutrophils upon activation by TNFα, GM-CSF, and PAF. This pathway leads to endothelial cell proliferation and differentiation. Meanwhile, the aforementioned TIMP-1-free MMP9 truncates the amino terminal of CXCL8, contributing to a positive feedback loop. Neutrophil-derived H_2_O_2_ activates Ets-1, which is responsible for the expression of MMPs. Additionally, the binding of ICAM-1 or E-selectin with neutrophils during rolling on the endothelium triggers intracellular signaling in endothelial cells [73].

## 5. Mechanism of Activation and Function of Neutrophils

Neutrophils perform a crucial role in initiating and regulating long-term anti-tumor immune signaling responses and participate in bidirectional interactions with other immune cells (B and T-cells, NK cells, and DCs) [74,75,76]. Due to their capacity to present tumor antigens [77] through RBC phagocytosis [78] and subsequent presentation through MHC and other costimulatory molecules, neutrophils play a crucial role in recruiting anti-tumor CD8+ effector T-cells, DC, macrophages, and NK cells in TME. TLR-stimulated neutrophils extracellular trap release [79], which can then activate plasmacytoid dendritic cells (pDCs) [80] and T-cells [81]. pDCs are specialized in producing type I interferons (IFN-α and IFN-β). pDCs can activate T-cells, which then produce cytokines to further activate T-cells [82,83]. Contrarily, GM-CSF, and INF-γ are necessary factors for the acquisition of features of antigen-presenting cells (APC) in neutrophils [78]. N1 subtype secretes CCL3, CCL9, CXCL10, TNF-α, IL12, Cathepsin G, and neutrophil elastase (NE) to recruit, activate and induce proliferation of T-cells to implement anti-tumor influence and improve adaptive immune responses [84,85].

CD11b+Ly6G+ neutrophils produce IL-17 in tumor-bearing mice which promotes tumor growth through an IL-6-Stat3 signaling pathway [86]. Accordingly, an increase in CD8+ T-cell response and reduction in tumor growth in the murine lung cancer system can be achieved by inhibition of IL17 production in CD11b+Ly6G+ neutrophils [86]. On the other hand, CD66b+ neutrophil infiltration is associated with CD8+ T-cell infiltration, colocalization and improved responsiveness of CD8+ T-cells to TCR stimulation [87] in colorectal cancer [88], indicating that these cells may be utilized for promoting antitumor immunity. Neutrophils can be activated by CD64, CD32a, CD16a via IgG-based antibodies. Mechanistically, like any other immune cell, the Fc receptors (FcR) of neutrophils interact with monoclonal antibodies (mAbs) by binding to their Fc domains. Interaction between neutrophil-FcR and Fc region of the antibody triggers antibody-dependent cellular cytotoxicity (ADCC), antibody-dependent cellular phagocytosis (ADCP), or trogocytosis and release of some tumoricidal mediators. Opsonization of malignant cells, which is an antibody-dependent direct killing mechanism in neutrophils is facilitated by monoclonal antibodies [88]. Neutrophils express different types of FcR for IgG(FcγR) having subtypes capable of inducing FcγRI (CD64), FcγRIIa (CD32a), FcγRIIb (CD32b), FcγRIIc (CD32c), FcγRIIIa (CD16a), and FcγRIIIb (CD16b) having different holds for IgG1, IgG2, and IgG4 [89] in IgG-based cancer therapies. FcγRIIa is the most abundant FcR (160,000 copies/cell) present on neutrophils and is largely responsible for the anticancer activity of IgG Abs. Each Fc receptor of neutrophils has the capacity to bind to IgG-opsonizing tumor cells and to contribute to ADCC operation [90]. Neutrophils express FcγRI after activation. FcαR1 (CD89) has superior anti-cancer potential in various cancer types than CD64, which has a high affinity for the IgA. Interaction of FcαRI with IgA monoclonal antibodies (mAbs) can attract neutrophils to tumor-infiltrating effector memory T-cells (TEMs). FcγRIIa and FcγRIIIb represent high and moderate affinity for IgG, respectively. FcγRIIIb is abundantly present on neutrophils, where it acts as functional cross-linkers for the escalation of susceptibility of lymphoma cells to anti-CD20-mediated apoptosis by facilitating the interaction between CD20 and anti-CD20 mAbs. Neutrophils activated with G-CSF or IFN-γ strongly upregulate CD64 and inhibit the expression of CD16b [67]. This variation in Fc receptor regulatory frameworks may affect the way antibodies direct neutrophil effector functions. 

## 6. Immune Suppression by Neutrophils and Therapeutic Possibilities

Immunosuppression is an important hallmark of cancer. Multiple reports suggest that neutrophils can suppress both the innate and adaptive immune response during cancer initiation, progression, and metastasis. Neutrophils utilize a variety of pathways leading to immunosuppression that can be targeted in order to divert their tumor-promoting functions to the anticipated anti-tumor functions.

Neutrophils release type 1 arginase (Arg I) via degranulation for degrading arginine [91] leading to inactivation of T-cells and eventual immune suppression [92]. L-Arginine is important for maintaining the activities of T-cells through the expression of T-cell co-receptor CD3ζ [93]. Arginine enters into the cells via transporters like CAT1 and CAT2 where it is metabolized to NO by NO synthase (NOS) [94] leading to the conversion of CD4^+^CD25^-^ cells into CD4^+^CD25^+^ T regulatory cells i.e., induced immune suppression [95]. Induction of Tregs (NO-Tregs) is independent of cGMP but depends on p53, IL-2, and OX40. Furthermore, ARG-1 activity is associated with polarized, protumoral M2 tumor-associated macrophages. ARG1-positive PMN-MDSCs had strongly reduced expression of granzyme B and Ki67 in cytotoxic T-cells suppressing its function [96]. Arginase inhibitors as single agents and in combination with immune checkpoint therapy are under clinical trial against solid tumors (NCT02903914). In fact, INCB001158 (an Arginase inhibitor) immunotherapy which has been reported to reverse immunosuppressive effects of neutrophils and myeloid-derived suppressor cells (MDSCs) is under phase II/phase III clinical study in advanced biliary tract cancer in combination with first-line chemotherapy i.e., gemcitabine/cisplatin regimen [97].

Noteworthy, the elevation of nitric oxide in the tumor microenvironment is multifactorial. Nitric oxide has biphasic roles in tumor immunity. High and low NO concentrations have opposite effects on tumor immune responses. On the other hand, long-term exposure to NO can lead to ROS generation and apoptosis in neutrophils via the mitochondrial death pathway [98] and NO may also enhance NET formation through PI3K.myeloperoxidase or neutrophil elastase [99].

Neutrophils activated by GM-CSF have PD-L1 on their surface, which blocks T-cell growth through the PD-1/PD-L1 pathway in gastric cancer and hepatocellular carcinoma. Cancer-associated fibroblasts (CAFs) also make neutrophils express PD-L1, which impairs T-cell function through IL6/Stat3 signaling. A new peptide vaccine based on Arginase-1 combined with anti-PD1 has been shown to enhance T-cell infiltration in tumors and reduce the suppression of tumor-educated myeloid cells and change the M1/M2 ratio [100].

TANs use mitochondrial fatty acid oxidation to maintain their NADPH level under low glucose and then produce ROS that impair lymphocyte functions [101]. Tumor cell-released autophagosomes (TRAPs) also influence neutrophils to produce ROS and undergo caspase-3-mediated apoptosis. These apoptotic neutrophils then suppress T-cell activation and proliferation [102]. Drugs like Apocynin, DPI or TAT peptide inhibitors can target NADPH oxidase family members in cancer treatment [103].

Tumor-secreted factors influence myelopoiesis and produce pathologically activated neutrophils (PMNs), also called PMN-MDSCs, from bone marrow precursors [104]. PMN-MDSCs die by ferroptosis in the tumor microenvironment and produce oxygenated lipids that suppress T-cells of human and mouse origin [105]. Ferroptosis has been linked to drug resistance [106], but the effect of these drugs on PMN-MDSCs is poorly studied.

TANs can suppress the immune system by activating TGFβ with their MMP9 secretion in colorectal cancer [107]. The mentioned pathway blocks T-cell proliferation in mice models. Blocking MMP2/MMP9 and TGFβ thereby can reduce tumor growth. Other neutrophil-derived factors, such as IL-10, CD39, CD73, indoleamine-deoxygenase, PTGS2, etc., also help neutrophils to suppress T-cells indicating that these factors may be potential targets for therapy. Moreover, NETs can lead to T-cells exhaustion by expression of PD-L1 [108]. This function of NETs has been observed in colorectal cancer patients during surgery, suggesting that anti-PD-L1 therapy can be useful in this setting.

## 7. Good Neutrophils in the Tumor: An Indication of Role Reversal?

Neutrophils can stimulate or suppress tumor growth [77]. During the early stages of cancer, TANs can activate cytotoxic T-cell in stage I/II either by direct priming or by secreting several cytokines in human lung cancer. Elevated IFN-γ, GM-CSF and reduction of Ikaros transcription factor promote TANs to acquire a hybrid state which has neutrophil and antigen-presenting capacity [109] even though neutrophils are not professional APC. Tumor-draining lymph nodes as well, play a crucial role in cancer progression. TANs transmigrate to lymph nodes where they modulate T-cell response in a stage-dependent manner. In the non-metastatic stage, neutrophils in lymph nodes acquire antigen-presenting phenotype (HLA-DR^+^CD80^+^CD86^+^ICAM1^+^PDL1^−^) and activate cytotoxic T Cells (CD27^+^Ki67^hi^PD-1^−^) to improve patient prognosis [110]. As tumors progress, neutrophils cease to respond in an anti-proliferative manner and display PDL1 which is driven by GM-CSF and STAT3, enriching immunosuppressive T-cells (CD27^+^Ki67^hi^PD1^+^) [110,111]. As a result of gaining this phenotype, TANs initiate tumor- promoting activities such as angiogenesis [112], vascular permeability, tumor cell dissemination to other organs and immune suppression. 

In contrast to the earlier mentioned immunosuppressive role of GM-CSF induced neutrophils in the lung cancer model, Granot and his co-workers demonstrated anti-metastatic activity of G-CSF activated neutrophils through ROS production in TME which were drawn by tumor-secreted CCL2. [113,114]. The extent of ROS generation in tumors appears to serve as an indication of whether the response is going to be pro- or anti-tumorigenic. Since ROS generation is correlated with the activation of Lyn and Akt, inhibitors of Akt pathway should be able to reduce G-CSF-induced neutrophil activities [115]. Accordingly, therapeutic regimens should be carefully designed. Interestingly, earlier mentioned DPI as well, can control ROS production thereby proving interconnecting mechanisms. IL-17 secreted by neutrophils enhances their antitumor potential through the production of MPO, ROS, Interferon (IFN), and TFN-related apoptosis-inducing ligands (TRAIL), defensins, and proteinases. The MPO found in NETs has the potential to kill melanoma cells and slow tumor growth. Azide, cyanide, or the addition of catalases inhibit MPO activity and prevent neutrophil-mediated tumor cell lysis [114] thereby indicating therapeutic approaches. Generation of ROS by the activities of earlier mentioned NADPH oxidase, also result in the production of H_2_O_2_ leading to neutrophil-mediated cytotoxicity through Transient Receptor Potential Cation Channel 2 (TRPM2) activation and subsequent chemokine production by monocytes [116]. 

IL-17 production by γδ T-cells has been also shown to be responsible for G-CSF-dependent expansion of neutrophils and N2 polarization in mammary tumors [117] indicating the differential role of IL-17 in neutrophils and γδ T-cells. However, this tumor-promoting role was observed in early stages in contrast to late stages when depletion of neutrophils did not affect metastasis again suggesting either role reversal of the same after significant progression of the disease or loss of signal transduction pathway molecules specifically involved in tumor-cell-neutrophil interactions. Noteworthy, in a path-breaking discovery, neutrophils have been shown to accumulate in premetastatic lungs and inhibit metastatic seeding in mammary tumor models of mice [114] supporting the above-mentioned role-reversal hypothesis. The evidence mentioned, also suggests the possibilities of successful detection of metastatic sites through neutrophil tracking and accordingly designing targeted delivery of drugs.

The phenotype and function of TANs are determined by the tumor microenvironment, which can induce different polarization states of neutrophils, ranging from N1 (anti-tumoral) to N2 (pro-tumoral). The balance between the pro-tumor and anti-tumor effects of neutrophils is influenced by several factors, including the tumor type, the stage of disease, and the patient’s immune status. In some cases, neutrophils may promote tumor growth, while in other cases they may have anti-tumor effects.

## 8. Neutrophil-Derived Exosomes: Emerging Players in Cancer Metastasis

Exosomes were first reported in 1987 during in vitro culture of sheep reticulocytes and these small vesicles were able to perform different membrane receptor activities including acetylcholinesterase, cytochalasin B binding (glucose transporter) nucleoside binding (i.e., nucleoside transporter), Na+-independent amino acid transport and the transferrin receptor [118]. Exosomes are formed by a double invagination of the plasma membrane (PM) and the formation of multivesicular bodies (MVB) which contain intraluminal vesicles (ILV). The ILVs sizing about 40–160 nm are then released in the surrounding tissue environment or in circulation through the fusion of the MVBs with the PM and eventual exocytosis [119] which are termed exosomes. Exosomes are usually referred to as cargo vesicles which can carry proteins, lipids, genetic material, RNAs or membrane receptors [120]. The composition of exosomes is dependent on the cell type and tissue from which it originates. However, the proteome profile showed that 72% of proteins are common to most exosomes and these proteins contribute to their biogenesis [121]. Exosomes execute their mode of action in two ways: either by fusion with the recipient cell, since they have a membrane surrounding the material they carry, or by communicating with the receptors present on neighboring or distant cells, whereby they can reprogram the recipient cell.

Exosomes may arise de novo inside cells or may be consumed by an endosome. During the invagination of the PM, proteins, lipids, DNA or RNA are incorporated inside the vesicular formation. These vesicles can be considered as mini parent cells which provide a novel way to communicate with neighboring or distant cells. Almost all types of cells can generate exosomes, such as epithelial, mesenchymal, stromal, lymphocytes, monocytes, macrophages, NK, dendritic, endothelial and tumor cells [122,123,124,125,126,127,128]. The preferred naming for exosomes is based on the cell origin. Recent studies have shown that neutrophils also release exosomes [128] and can play a role independently outside neutrophils and orchestrate adaptive immune responses by influencing some cell types [14]. Exosomes possess functional resemblance to their parental cells as they do carry a partial cellular component of parental cells. However, they do not reconstitute parental cell [129].

## 9. Unravelling the Intricate Role of Neutrophil-Derived Exosomes in Cancer Progression

Neutrophil-derived exosomes (NDEs) play important roles in a variety of physiological and pathological processes, including inflammation, wound healing, and cancer [130,131]. In inflammation, NDEs can promote the recruitment of other immune cells to the site of inflammation where they release pro-inflammatory cytokines [132]. In wound healing, NDE promotes the proliferation and migration of fibroblasts. These migrated fibroblasts release growth factors that stimulate the growth of new tissue [133]. In the context of cancer, NDEs have been shown to play a role in promoting tumor growth and metastasis, as well as suppressing the immune response to cancer cells. However, NDEs have been rarely studied for tumor treatment [20]. Recently Genschmer et al., showed that NDEs contain NE on their surface. These NE-bound NDEs bind and degrade ECM via the integrin Mac-1. NE has been found to be associated with metastasis of non-small lung cancer into the aorta [134]. Similarly, in prostate and breast cancer NE activates MAPK to induce ERK phosphorylation enhancing tumorigenesis. The differential polarization of neutrophils also leads to skewed production of exosomes. N1 neutrophils produce N1 NDEs, whose molecular contents are biased towards inflammatory biomolecules, whereas N2 neutrophils produce N2 NDEs, whose molecular contents are biased towards regulatory biomolecules. N1 NDEs promote macrophage activation and T-cell proliferation, whereas N2 NDEs promote M2 polarization of macrophage and also reactivate tumor cells [135]. Proteases derived from NDEs play an important role in chronic inflammatory diseases of the lung, such as chronic obstructive pulmonary disease (COPD) and acute respiratory distress syndrome (ARDS), by contributing to tissue damage, mucus hypersecretion, and impaired host defense [136,137]. N2 NDEs also contain IL6 which promotes neutrophil polarization to N2 via the IL-6-STAT3-ERK1/2 signaling [138]. Conversely, N1 NDEs transport the Defensin 1 that recruits lymphocytes. N1 NDEs transport miR-223 which plays a critical role in the development of the myeloid lineage cells, promote immune cell recruitment and carry factors such as RAP1A which is known to prevent metastasis. N1 NDEs can also carry pro-inflammatory cytokines, such as IL-1β, IL-2 and IL-4. Recently an interesting observation was made by Kolonics et al. which suggested that neutrophils produce EVs based on the surrounding environment which might function as an indicator of tumor progression. When N1 neutrophils enter the TME they release N1 NDEs with anti-tumor signals, but as the tumor grows and escapes immune surveillance, the same reprograms the immune cells around itself. At this point, N1 neutrophils are switched to N2 phenotype and release N2 NDEs which promote tumor growth [135].

Although NDEs have been used previously to treat arthritis and sepsis, their role in cancer treatment remains unexplored mainly due to the low yield of NDEs and the short half-life of neutrophils themselves. 

## 10. NDEs: Driving Tumor Invasion and Metastasis

Cancer metastasis consists of sequential steps of digestion of extracellular matrix (ECM) and dissemination of tumor cells from the primary tumor, intravasation, transportation to a suitable niche, extravasation and adaptation to the secondary site and subsequent proliferation. The mentioned steps require the secretion of matrix metalloproteinases for dissolving the ECM, manipulation of adhesion molecules and transportation. NDEs are extracellular vesicles that can influence cancer progression and metastasis by carrying various bioactive molecules. For example, Zhu et al. (2022) showed that these exosomes carried miR-23a-3p that targeted PTEN and enhanced liver cancer cell invasion [20]. Wang et al. (2019) demonstrated that NDEs carried S100A8/A9 that facilitated lung metastasis by creating pre-metastatic niches [139]. Park et al. (2019) revealed that NDEs carried CD177 that induced regulatory T-cell differentiation and suppressed anti-tumor immunity [140]. Tian et al. (2018) reported that NDEs carried MMP9 that stimulated angiogenesis and increased gastric cancer cell invasion and metastasis [141]. 

Importantly, exosomes derived from the ascites of patients with ovarian cancer patients, activated CD8+ T-cells [142] and Treg cells [143] secrete MMP9. Exosomes derived from tumor cells, mesenchymal stem cells, Treg cells [144], and mast cells are known to secrete VEGF [145]. Cathepsin G is secreted by neutrophils and other immune cells. It has been reported that cathepsin G can be released by exosomes. Cathepsin G may promote angiogenesis, the formation of new blood vessels by cleaving and activating VEGF [146]. Cathepsin G also induces the expression of MMPs [147]. Cathepsin G has been shown to stimulate TGF-β signaling for upregulating VEGF and MCP-1 [148]. Therefore, analysis of neutrophil-derived exosomes for the presence of these factors may serve in understanding disease prognosis and open new avenues in disease treatment. These studies highlight the diverse roles of NDEs in promoting cancer metastasis through different mechanisms.

## 11. NDEs as Therapeutic Tools in Cancer

Understanding the role of NDEs in tumor progression has the potential to lead to new treatments that can target these processes and improve outcomes for cancer patients. One promising approach is to develop therapies that can inhibit the production or function of N2 NDEs, preventing them from promoting immune system suppression and tumor growth. Another approach is to use NDEs as a delivery system for cancer treatments. NDEs can transport molecules directly to cancer cells, allowing for targeted and efficient delivery of cancer therapies. Additionally, NDEs can be engineered to target specific cancer cells, further increasing their effectiveness as a delivery system for cancer treatments. In fact, NDEs have been decorated with superparamagnetic iron oxide nanoparticles (SPIONs) to achieve higher tumor-targeting therapeutic effects [20]. Some potential approaches (Figure 3) that are currently being investigated to treat solid cancers are discussed below.

### 11.1. Inhibition of Exosome Release

Exosome biogenesis and release inhibitors are a new class of cancer therapies that aim to block the production or release of exosomes from cancer cells or other tumor-supporting cells such as N2 neutrophils. N2 TANs-derived exosomes can carry tumor-promoting factors, such as growth factors and cytokines, to other cells and stimulate angiogenesis, invasion, and metastasis [149]. Additionally, NDEs can suppress the immune system by inducing apoptosis of T-cells, inhibiting NK cell activity, and promoting the expansion of regulatory T-cells and myeloid-derived suppressor cells [150]. Therefore, blocking N2-TANs-derived exosome and tumor cell-derived exosome is a rational strategy to inhibit tumor progression. Exosome biogenesis and release inhibitors can interfere with the endosomal sorting complex required for transport (ESCRT) machinery or other proteins involved in exosome biogenesis and secretion, such as TSG101, CD9, Alix, SMase2, Rab11, ARF6, and Rab27 [151,152] which can help to slow down or stop tumor growth. Rab27a and b are involved in the docking and fusion of neutrophil granules with the plasma membrane, which is an essential step in the release of neutrophil degranulation enzymes into the extracellular environment. A recent study has demonstrated that the knocking down of Rab27a can inhibit exosome secretion [153]. In the case of Rab27b, more research is needed to investigate the potential role of Rab27b overexpression in NDEs in the context of tumor growth and treatment. In melanoma, Rab27a overexpression is correlated with decreased patient survival [154]. Blockage of the ARF6-based pathway causes mitochondrial aggregation near the microtubule-organizing center and subsequently induces detrimental reactive oxygen species (ROS) production [155]. Inhibition of exosome release by tumor cells can inhibit NET formation and release of mature N2 neutrophils at the tumor site. Some of the most promising exosome biogenesis and release inhibitors include Manumycin A, Tipifarnib, Macitentan, Nexinhib20, ARF6 inhibitors, etc. Manumycin A suppresses Ras/Raf/ERK1/2 signaling and hnRNP H1, a splicing factor that regulates exosome cargo [156]. Tipifarnib reduces ESCRT-0 proteins HRS and ALIX, as well as Rab27a [151]. Macitentan blocks the activation of endothelin receptors and downstream signaling pathways, such as ERK1/2 and Akt [152]. Nexinhib20 reduces ceramide levels and prevents MVB formation and ARF6 inhibitors impair the fusion of MVBs with the plasma membrane.

### 11.2. Modulation of Exosome Content

Modulation of the content of NDEs can be a potential strategy to target cancer. NDEs contain various bioactive molecules, such as microRNAs (miRNAs), proteins, and lipids, which can promote or inhibit cancer growth and metastasis. By modulating the content of NDEs, it may be possible to alter their effect on cancer cells and enhance their therapeutic potential. NDEs can be loaded with tumor-suppressive molecules such as miRNAs [157] that target oncogenes and promote tumor suppressor pathways. For example, Block the release of pro-tumorigenic molecules such as TGF-β [158] by NDEs during advanced stages of cancer, could help to reduce the pro-tumorigenic effects of NDEs on cancer cells [159]. Furthermore, engineering NDEs to express specific proteins or peptides that target cancer cells, or their microenvironment could be another strategy to enhance their therapeutic potential. For instance, NDEs could be decorated with specific ligands that bind to receptors on cancer cells, which can increase their uptake by cancer cells and improve their efficacy as a delivery system for cancer treatments. 

### 11.3. Neutrophil-Derived Exosomes

Targeting NDEs is a promising approach for cancer therapy. This can be achieved by using the antibodies that recognize and bind to specific surface proteins on NDEs, such as CD63, CD9 and CD81 [160,161,162,163]. LFA-1 (lymphocyte function-associated antigen-1) is another receptor present on the surface of neutrophils and plays a role in neutrophil adhesion and migration [164]. The mentioned antibodies can be conjugated to drugs or other therapeutic agents, allowing them to be delivered specifically to cancer cells via N2 NDEs. Nanoparticles, including liposomes and gold nanoparticles, can also serve as carriers for drug delivery or other therapeutic agents to NDEs. These nanoparticles can be engineered to target NDEs specifically and coat them with molecules that bind to surface proteins on NDEs. Once taken up by NDEs, the nanoparticles can deliver their cargo specifically to cancer cells, resulting in the targeted and efficient delivery of cancer therapies.

### 11.4. Usage of Targeted Antibodies

Antibodies are promising new therapeutic agents for cancer, and they are being investigated in a number of clinical trials [165]. An antibody that targets a cancer cell surface protein can be used to activate T-cells, which can then attack and kill the cancer cells. There are just a few studies that have been done on the use of antibodies in NDEs to fight cancer and inflammation. Some of the specific antibodies that are being used in NDEs are Anti-EGFR, PD1 and CTLA-4. EGFR is a protein that is often overexpressed in cancer cells. Anti-EGFR antibodies can be used to target NDEs to cancer cells, or they can be used to deliver drugs or other therapeutics to cancer cells [166]. PD-1 (Programmed cell death protein 1) is expressed by T Cells and helps to regulate the immune response. When PD-1 binds to its ligands, PD-L1 or PD-L2, which are often overexpressed in cancer cells [167], it can inhibit T-cell activation and promote immune evasion by cancer cells. pembrolizumab and nivolumab, have been developed to block the interaction between PD-1 and its ligands, thereby releasing the brakes on the immune system and allowing T-cells to attack cancer cells [168]. These antibodies have been shown to be effective in the treatment of several types of cancer, including melanoma, non-small cell lung cancer, and bladder cancer. Another receptor that is expressed by T-cells acts as a negative regulator for T-cell activation and proliferation by binding to its ligand B7-1 and B7-2 present in cancer cells. Ipilimumab, an anti-CTLA-4 antibody, can block the interaction between CTLA-4 and its ligands, thereby enhancing T-cell activation and promoting an immune response [169].

### 11.5. Engineer NDEs to Express Anti-Cancer Proteins

NDEs can be engineered to express genes that encode for anti-cancer proteins such as TRAIL protein which can be used to kill cancer cells. TRAIL binds to cell surface death receptors, TRAIL-R1 (DR4) and TRAIL-R2 (DR5) and facilitates the formation of a death-inducing signaling complex (DISC), eventually activating the p53-independent apoptotic cascade [170]. This unique mechanism makes TRAIL a potential anticancer therapeutic, especially for p53-mutated tumors. Further research is needed to determine the feasibility and efficacy of engineering NDEs to express anti-cancer proteins such as TRAIL. While this approach holds promise, there are still significant challenges to be addressed, including optimizing the delivery and targeting of these therapeutic agents. In an exceptional study, carfilzomib, a second-generation proteasome inhibitor, was loaded into nitrogen cavitation (NC)-derived extracellular vehicles (EVs) which efficiently depleted circulating tumor cells (CTCs) and inhibited metastasis in murine models [171]. NC instantly disrupts neutrophils to form nanosized membrane vesicles. NC-derived EVs are similar to naturally secreted EVs by neutrophils but unlike natural EVs they include less nuclear acids and organelles. However, NC enhanced EV production by 16 times, opening up an avenue for achieving higher production of EVs for eventual encapsulation. Eventually, NC-derived EVs loaded with the anti-inflammatory drug piceatannol inhibited lung inflammation and sepsis induced by lipopolysaccharide (ref). The above-mentioned approach may be also used to deliver siRNA-based cancer vaccines given the ability of NDEs to accumulate at the tumor site in contrast to nanoparticles which have the problem of accumulating in the liver. Further research is needed to investigate the feasibility and safety of these approaches and to identify the most effective cargo for NDEs in cancer therapy.

### 11.6. Immune Modulation

Neutrophil-derived exosomes can modulate the immune response, and this property can be utilized to target solid cancers. NDEs have been shown to have immunomodulatory properties that can be harnessed for cancer therapy. One approach involves loading NDEs with immune-stimulatory molecules, such as toll-like receptor (TLR) ligands [172], to activate the immune response against cancer cells. TLR ligands can activate immune cells by binding to TLRs, which are expressed on the surface of various immune cells, including dendritic cells and T-cells or in intracellular compartments such as the endosome, ER, endosome, and lysosome. By loading NDEs with TLR ligands, it is possible to stimulate the immune response and enhance anti-tumor immunity. NDEs loaded with a TLR9 ligand could activate dendritic cells and induce anti-tumor immunity in a mouse model of melanoma [173]. Furthermore, the TLR4 ligand and TLR1/2 ligand can bind to exosomal surfaces. Consequently, these exosomes can activate bystander DCs and NK cells, augment secretion of proinflammatory and immunoregulatory cytokines and mediate enhanced Th1 polarization [174]. Another example is CD11b, which is a surface receptor integrin on neutrophils. It has been shown that CD11b activation can promote anti-tumor immunity by stimulating pro-inflammatory macrophage polarization [175] and recruitment to the tumor microenvironment and their interaction with cancer cells [31]. 

### 11.7. Combination Therapy

Combining different strategies has been shown to enhance the efficacy of neutrophil-derived exosome-based therapies. neutrophil-derived exosomes were loaded with the chemotherapy drug doxorubicin and a small interfering RNA (siRNA) targeting the anti-apoptotic protein Bcl-2, as well as immune-stimulatory CpG oligonucleotides. In a mouse model of melanoma, these exosomes induced apoptosis and enhanced anti-tumor immunity, resulting in a significant reduction in tumor growth [176]. The combination of doxorubicin and Bcl-2-targeting siRNA enhances the cytotoxic effects of chemotherapy and promotes apoptosis in tumor cells [177]. Additionally, the inclusion of CpG oligonucleotides in the exosome cargo can stimulate the immune system, leading to an increased anti-tumor response [178]. Neutrophil-derived exosomes are an attractive therapeutic option due to their ability to target cancer cells selectively and efficiently [20].

Overall, the role of neutrophil-derived exosomes in therapy is still in the early stages of investigation. However, their potential therapeutic applications are promising and warrant further research to fully understand their mechanisms and potential clinical applications.

## 12. Current Advancements and Future Perspectives

Numerous evidence supports the pro-tumor role of N2 neutrophils but targeting them remains a challenge due to the lack of clear molecular discrimination between TANs, especially PMN-MDSCs, from the remaining neutrophil populations. Technological advances, especially in single-cell profiling techniques, such as single-cell proteomics and single-cell RNA-seq, may greatly enhance the abilities to classify heterogeneous neutrophil populations into “bad neutrophils” or N2 neutrophils, which can be targeted for therapy, and “good neutrophils” or N1 neutrophils, which need to be spared to prevent neutropenia which is a major side-effect of neutrophil-targeted therapy as well as chemotherapy. Therefore, it is important to standardize the time and dosage of G-CSF administration for optimal recovery. Moreover, the effects of G-CSF administration on N1 and N2 subpopulations should be investigated and documented.

The mentioned roles that neutrophils take on when dealing with cancer emphasize their adaptability and capacity to react to a wide range of targets including modified self-cells through pattern recognition molecules (PRMs) such as collectins, picolins and pentraxins [179,180] both inside and outside the TME. At an early stage before metastasis, neutrophils have been already shown to transmigrate to lymph nodes for anti-tumor T-cell priming, which is converted into immunosuppressive functions after stimulation by GM-CSF/STAT3 [181]. Therapies aimed to target N2 neutrophils are being assessed in clinical trials. CD33+ in combination with HLA^-^DR^−^ CD15^+^ is used to identify N2 neutrophils. A novel tetravalent bispecific antibody AMV564 targeting CD33/CD3 has been developed and is currently being evaluated in early-phase clinical trials (NCT03144245) for relapsed or refractory acute myeloid leukemia (AML). The efficacy of LXRβ agonists, such as GW3965 and RGX-104, for the depletion of pro-tumor PMN-MDSCs are also being tested. Clinical trials determining the efficacy of blocking PMN-MDSC recruitment in enhancing cancer immunotherapy (NCT03161431) are underway. 

In addition to research focusing towards selectively targeting the N2 neutrophils, the population as a whole is being considered to carry anti-tumor drugs owing to their mobility and responsiveness. A study by Wang et al., demonstrated that activated neutrophils could uptake drug-loaded albumin nanoparticles (NP) via Fcγ receptors. Chu et al. showed that NP can be efficiently delivered across the blood to the tumor tissue via neutrophils after induction of inflammation. Neutrophils have been shown to efficiently deliver anti-CD11b antibody-linked gold nanorods or pyropheophorbide-a-loaded NPs in tumor cells inducing anti-tumor effects [171]. Noteworthy, acute stimulation of inflammation is required for neutrophil recruitment, either via monoclonal antibody or photosensitization. Neutrophils can also efficiently deliver drugs across the blood–brain barrier (BBB) to inflamed brain tumors. For example, neutrophils have been reported to deliver paclitaxel (PTX) containing liposomes to the brain and prevent the recurrence of glioma in murine models after surgical resection of tumors and thereby improving survival. Xue et al. showed that PTX liposomes exert negligible toxicity on neutrophils [171] High concentration of inflammatory signals triggered infiltration of neutrophils and the release of liposomal PTX. However, the delivery of drugs via neutrophils remains challenging owing to the short half-life, and anti-cancer drugs such as PTX can significantly affect and impair neutrophil functions, viability, and migratory potential.

Evidence demonstrated that under the influence of the external magnetic field, superparamagnetic iron oxide nanoparticles (SPIONs) modified by NDE loaded with the chemotherapy drug doxorubicin (DOX) can be highly targeted and enriched at tumor sites, inhibiting tumor growth to a significant degree [11]. Intravenous injection of DOX-loaded NDEs suppressed tumor growth and increased survival time in murine models [20]. In another interesting study, Gao et al. demonstrated a unique strategy to generate extracellular vehicles (EVs) using nitrogen cavitation (NC)which instantly disrupts neutrophils to form nanosized membrane vesicles.

Overall, despite significant advances in understanding the characteristics of TANs, their importance in both tumor promotion and anti-tumor functions and therapeutic approaches targeting N2 neutrophils, implementation of the findings still need clinical adaptability. In this context, the identification of N1 and N2 NDE contents, the relevance of the NDE contents with the progression of the disease and further exploration regarding planned delivery systems of NDE-based nanotherapeutics seem extremely relevant and necessary for future research.

## Figures and Tables

**Figure 1 vaccines-11-01028-f001:**
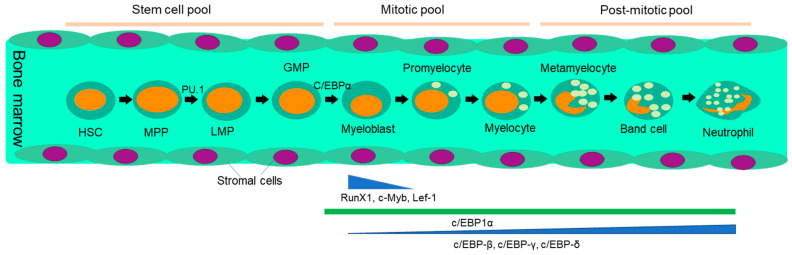
Schematics show the developmental stages of neutrophils. Neutrophil development in bone marrow occurs in three phases: the stem cell pool, the mitotic pool, and the post-mitotic pool. The stem cell pool includes the hematopoietic stem cell (HSC) to the granulocyte and macrophage precursor (GMP). The mitotic pool holds the myeloblast, promyelocyte, and myelocyte stages. The third phase, the post-mitotic pool, covers the metamyelocyte to mature neutrophil stages. Different transcription factors come into play at different stages of neutrophil development. PU.1 promotes MPP conversion to LMP, while Runx1 and c-myb stimulate myelocyte proliferation and azurophilic granule synthesis. The downregulation of Runx1 and c-Myb promotes myeloblast to promyeloblast conversion. Only C/EBPα remains constant throughout the subsequent differentiation stages of neutrophils. c/EBP-β, c/EBP-γ, and c/EBP-δ levels increase as terminal differentiation proceeds. Stromal cells secrete ligands like VCAM-1 and CXCL-12 specific to the receptor to be displayed by mature neutrophils. Further developmental stages of neutrophils include the myeloblast, promyelocyte, myelocyte, metamyelocyte, and band cell stages. Finally, band cells are converted into mature neutrophils, which are released into circulation.

**Figure 2 vaccines-11-01028-f002:**
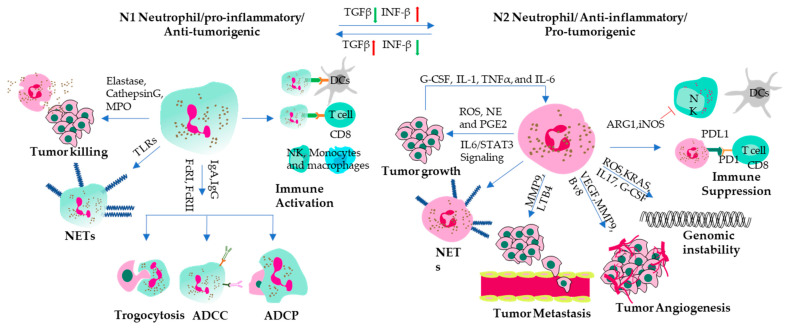
Anti-tumor and pro-tumor activities of neutrophils. The N1 subtype helps to generate an anti-tumorigenic niche. These cells can kill tumor cells directly by secreting elastase, cathepsin G, and myeloperoxidase. They can also act as non-professional APCs to activate other immune cells. TLR-stimulated neutrophils generate neutrophil extracellular traps (NETs). FcR I and FcR II can bind to the Fc region of mAbs and promote antibody-dependent cellular cytotoxicity, phagocytosis, and trogocytosis of tumor cells. On the other hand, N2-TANs influence several mechanisms that promote tumor spread. These cells release MMP-9, which frees VEGF from the ECM and stimulates angiogenesis. N2 TANs also secrete cytokines (IL-1β, TNF-α, IL-6, and IL-12) that cause chronic inflammation and arginase 1, which blocks CD8+ T-cells, prevents NK cells and APCs from reaching the tumor site or presenting antigens to cytotoxic T-cells, creating an immunosuppressive environment. Moreover, they produce ROS that can cause DNA damage and genotoxic effects on tumor cells. Tumor cells also secrete G-CSF, IL-1, TNF-α, and IL-6, which promote the differentiation of N2 subtype neutrophils. The LTB4 and MMP9 secreted by neutrophils support metastasis-initiating cell survival and bring the tumor cells into circulation.

**Figure 3 vaccines-11-01028-f003:**
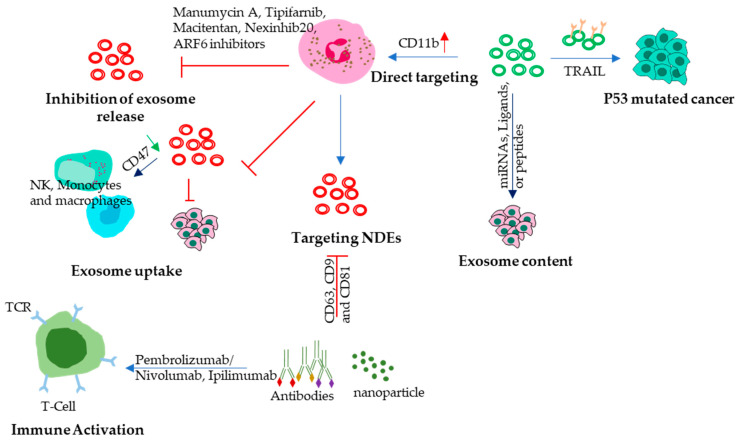
Strategies to use NDEs as therapeutic tools. Tumor cells, N2 TANs, and TANs-derived NDEs can be targeted in several ways. Exosome release can be inhibited using different drugs that are involved in their formation and secretion, by modulating their uptake by tumor cells, or by inducing uptake by phagocytic cells. NDEs can be used to deliver miRNAs to target oncogenic proteins such as TGFβ in advanced stages of cancer. N2 TANs NDEs can also be directly targeted using drug-loaded antibodies or nanoparticles against CD63, CD9, CD81, and LFA1. Using specific antibodies such as anti-PD-1 and CTLA4 can help in releasing the brakes on immune suppression. Tumors with mutated p53 can be targeted using Trail receptor-loaded NDEs.

## Data Availability

Not applicable.

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
