# Peer review of "Neutrophils in Cancer and Potential Therapeutic Strategies Using Neutrophil-Derived Exosomes"

_vaccines, 2023, doi:10.3390/vaccines11061028_

Round 1

Reviewer 1 Report

The paper is a long and complex review of a new particular topic.  I would have expected that authors discuss how a high rate of circulating lymphocytes/neutrophils in lung cancer is a clinically good prognostic indicator.  Besides, in the histology of carcinoid tumors, tissue infiltration by lymphocytes indicates a more favorable prognosis. [See:  Am J Clin Oncol  2018, 41, 1288; Histopathology 2017,70,1175; Chest 2017,151, 1186.]. Besides, some other flowcharts can allow a better comprehension of the text.  

Author Response

Responses to reviewer

The paper is a long and complex review of a new particular topic.  I would have expected that authors discuss how a high rate of circulating lymphocytes/neutrophils in lung cancer is a clinically good prognostic indicator. Besides, in the histology of carcinoid tumors, tissue infiltration by lymphocytes indicates a more favorable prognosis. [See:  Am J Clin Oncol 2018, 41, 1288; Histopathology 2017,70,1175; Chest 2017,151, 1186.]. Besides, some other flowcharts can allow better comprehension of the text. 

The reviewer is correct about the paper being long and complex. It is a review of a new topic that is rarely discussed, and we therefore wanted to cover it as completely as we could. We appreciate that the reviewer read the paper carefully and thought about how it could be improved. Regarding circulating neutrophils, we have already mentioned a few points regarding prognosis throughout the paper.

Notably, immune score has been correlated with good prognosis in neuroendocrine carcinoid tumor [1-4]. Immune score is calculated based on the number of immune cells infiltrating the tumor, expression of a panel of genes involved in immune response and the level of inflammation. Most parameters are dependent on age, so reference ranges are classified in four age groups i.e., children, adolescents, adults, and elders. The number of CD4+, CD8+ T cells, B cells but not NK cells, are negatively correlated; however, functionally they are positively correlated with age [5]. High immune score represents better response of immune system to fight with cancer than with low immune score.  Although neutrophils are major constituent of lymphocytes, high densities of intratumoral neutrophils have shown independent association with short survival and other localized neutrophils i.e., peritumoral and stromal were having no association [6]. The high intratumoral neutrophil to lymphocyte ratio (NLR) > 4 is correlated with poor survival and prognosis similar to macrophage to lymphocyte (MLR) ratio. Therefore, NLR correlation with prognosis is a misleading term because neutrophils are double edged sword and their phenotype strictly depend on the location of neutrophils, tumor niche, stage, and type of tumor. 

However, we could not include flowcharts at this time since it would require extensive modification of the whole manuscript which needs further time. But we thank you for the suggestion of including flow charts. This will help us improve our future manuscripts to attract more readers.

References

[1] Roncati L, Manenti A, Piscioli F, Pusiol T, Barbolini G. The Immune Score as a Further Prognostic Indicator in Carcinoid Tumors. Chest. 2017;151:1186.

[2] Wolin EM. Advances in the Diagnosis and Management of Well-Differentiated and Intermediate-Differentiated Neuroendocrine Tumors of the Lung. Chest. 2017;151:1141-6.

[3] Roncati L, Manenti A, Piscioli F. Upgrading the Diagnostic Assessment in Primary Pulmonary Carcinoids: The Tumor-infiltrating Lymphocytes (TILs). Am J Clin Oncol. 2018;41:1288-9.

[4] Roncati L, Manenti A, Piscioli F, Pusiol T, Barbolini G. Immunoscoring the lymphocytic infiltration in carcinoid tumours. Histopathology. 2017;70:1175-7.

[5] Tang G, Yuan X, Luo Y, Lin Q, Chen Z, Xing X, et al. Establishing immune scoring model based on combination of the number, function, and phenotype of lymphocytes. Aging (Albany NY). 2020;12:9328-43.

[6] Ocana A, Nieto-Jimenez C, Pandiella A, Templeton AJ. Neutrophils in cancer: prognostic role and therapeutic strategies. Mol Cancer. 2017;16:137.

Reviewer 2 Report

In this review, Neutrophils in Cancer and Potential Therapeutic Strategies us- 2

ing Neutrophil-derived Exosomes'' This review will focus on the communication pathways between tumors and 28 neutrophils, and the role of neutrophil-derived exosomes (NDEs) in tumor growth.

Even though including recent and relevant literature, could be structured more clearly and follow a line of thought.

Comments:

1. Neutrophils are the most abundant immune cells and make up about 70% of white blood cells in human blood and play a critical role as the first line of defense in the innate immune response. However, with the rise of interest in T-cell biology, and since no clear roles for neutrophils in cancer were defined, a gradual decrease in the interest in these major cells of the immune system was noted in the following decade, even as new immunotherapy modalities were developing. Interest has increased during the last years, as recent data suggest more important and significant roles for neutrophils in tumor biology than previously thought. More clarify the exact mechanism of neutrophils association with T-cells in close association with tumor cells and within tumor vasculature should be addressed.

2. Increase in circulating neutrophils has been described in several animal models of cancer, such the 4T1 mammary tumor, Lewis Lung Carcinoma mice model, and the 13762NF mammary adenocarcinoma rat model. The finding of the existence of circulating LDN and HDN in cancer has raised new questions regarding the origin for these subsets also need to be attention in this review.

3. Although the cause for the presence of mature and immature LDN in cancer is still unclear,it ie important to mention the isolation of the neutrophils together with the PBMC fraction could result from (a) different degrees of maturation and (b) different activation and degranulation states. Also the fact that newly formed neutrophils can be seen in the LD fraction at an earlier time point than the HDN and more connection should suggest that these are immature cells rapidly leaving the bone marrow.

4. Possible sources of the murine cancer-related neutrophil subsets, in the circulation and the tumor tissue. The source of the different cancer-related neutrophil subtypes is still unclear. The immature subset of the LDN could originate from the bone marrow or spleen as immature neutrophils that fail to undergo terminal differentiation.

5. Since chemotherapy causes neutropenia, neutrophil count during treatment is an important factor in the decision making of duration and dosage of treatment. It was shown that neutropenia following chemotherapy can be countered by administering recombinant G-CSF. In this review, the activation of neutrophils with GM-CSF should be addressed.

Author Response

Responses to reviewer

In this review, Neutrophils in Cancer and Potential Therapeutic Strategies using Neutrophil-derived Exosomes'' This review will focus on the communication pathways between tumors and 28 neutrophils, and the role of neutrophil-derived exosomes (NDEs) in tumor growth.

Even though including recent and relevant literature, could be structured more clearly and follow a line of thought.

Comments:

  1. Neutrophils are the most abundant immune cells and make up about 70% of white blood cells in human blood and play a critical role as the first line of defense in the innate immune response. However, with the rise of interest in T-cell biology, and since no clear roles for neutrophils in cancer were defined, a gradual decrease in the interest in these major cells of the immune system was noted in the following decade, even as new immunotherapy modalities were developing. Interest has increased during the last years, as recent data suggest more important and significant roles for neutrophils in tumor biology than previously thought. More clarify the exact mechanism of neutrophils association with T-cells in close association with tumor cells and within tumor vasculature should be addressed.

We thank the reviewer for highlighting the need to focus on this point. We would like to draw the attention of the esteemed reviewer towards the already discussed mechanisms of association between the neutrophils and the T cells under the tumor microenvironment in Section 5 and Section 6, where both the tumor-promoting and tumor suppressing interactions of TANs with T cells have been elaborated with references. In this regard, we have included a few more points to clarify things. The association with tumor vasculature is discussed separately in Section 4.  

  1. Increase in circulating neutrophils has been described in several animal models of cancer, such the 4T1 mammary tumor, Lewis Lung Carcinoma mice model, and the 13762NF mammary adenocarcinoma rat model. The finding of the existence of circulating LDN and HDN in cancer has raised new questions regarding the origin for these subsets also need to be attention in this review.

We are grateful for pointing out the need to clarify the terminologies N1/N2 vs. LDNs and HDNs along with their origins. Accordingly, we have added a paragraph at the end of Section 3. In the following sections, we discuss the conversion of N1/N2 during different signaling events, which is also related to the origins.

  1. Although the cause for the presence of mature and immature LDN in cancer is still unclear,it ie important to mention the isolation of the neutrophils together with the PBMC fraction could result from (a) different degrees of maturation and (b) different activation and degranulation states. Also the fact that newly formed neutrophils can be seen in the LD fraction at an earlier time point than the HDN and more connection should suggest that these are immature cells rapidly leaving the bone marrow.

The suggestion is well taken into consideration. We have added a few points at the end of Section 3 regarding factors contributing towards immature LDNs. Furthermore, with the reviewer's permission, we have also incorporated the mentioned comment into the same paragraph.

  1. Possible sources of murine cancer-related neutrophil subsets, in the circulation and the tumor tissue. The source of the different cancer-related neutrophil subtypes is still unclear. The immature subset of the LDN could originate from the bone marrow or spleen as immature neutrophils that fail to undergo terminal differentiation.

The added portion of Section 3 discusses the different sources of neutrophils in humans and mice, the necessity of analyzing subpopulations in cancer scenarios and the sources of mobilization and the differences in surface markers. We have also provided evidence where the transcription factors responsible for terminal differentiation of neutrophils have been identified. Additionally, we have discussed the cancer-associated immature neutrophil subpopulations of mice in terms of isolation source.  

  1. Since chemotherapy causes neutropenia, neutrophil count during treatment is an important factor in the decision making of duration and dosage of treatment. It was shown that neutropenia following chemotherapy can be countered by administering recombinant G-CSF. In this review, the activation of neutrophils with GM-CSF should be addressed.

We would like to draw the valued reviewer's attention towards Section 1, Section 2, Section 3, Section 5 and Section 7, where G-CSF-induced neutrophil activations have been discussed. Furthermore, we have added this valuable comment to both Section 2 and Section 12 in response to your recommendation.

Reviewer 3 Report

The present manuscript describes “Neutrophils in Cancer and Potential Therapeutic Strategies using Neutrophil-derived Exosomes (NDEs)”. Neutrophil derived exosomes are known to induce anti-tumor effects via activation of the apoptotic signaling pathway. NDEs can efficiently cross blood brain barrier (BBB) proving to be an effective therapeutic approach to target inflamed brain tumors. The authors have described comprehensive review about neutrophils. The authors have covered how topics about generation of neutrophils to tumor infiltration, tumor associated neutrophils subdivisions into tumor-associated macrophages, intra-tumoral activities of neutrophils, mechanism of activation and function of neutrophils, immune suppression by neutrophils and therapeutic possibilities, how neutrophil-derived exosomes emerged players in cancer metastasis, Unravelling the intricate role of neutrophil-derived exosomes in cancer progression, driving tumor invasion and metastasis, therapeutic tools in cancer, current advancements and future perspectives. The authors have provided complete review of neutrophils-derived exosomes as therapeutic agents for cancer. I would recommend this article to publish in Vaccines Journal.

Author Response

Responses to reviewer

The present manuscript describes “Neutrophils in Cancer and Potential Therapeutic Strategies using Neutrophil-derived Exosomes (NDEs)”. Neutrophil derived exosomes are known to induce anti-tumor effects via activation of the apoptotic signaling pathway. NDEs can efficiently cross blood brain barrier (BBB) proving to be an effective therapeutic approach to target inflamed brain tumors. The authors have described comprehensive reviews about neutrophils. The authors have covered how topics about generation of neutrophils to tumor infiltration, tumor associated neutrophils subdivisions into tumor-associated macrophages, intra-tumoral activities of neutrophils, mechanism of activation and function of neutrophils, immune suppression by neutrophils and therapeutic possibilities, how neutrophil-derived exosomes emerged players in cancer metastasis, Unravelling the intricate role of neutrophil-derived exosomes in cancer progression, driving tumor invasion and metastasis, therapeutic tools in cancer, current advancements and future perspectives. The authors have provided complete review of neutrophils-derived exosomes as therapeutic agents for cancer. I would recommend this article to publish in Vaccines Journal.

We would like to express our gratitude for recommending the manuscript for publication in the Vaccines journal.

Reviewer 4 Report

This review article by Abhishek Dutta et. Al. is well presented but some concerns are addressed below.

1.       Line 231: The size of neutrophils needs to be clearly defined. Neutrophils once entering tissues and attached to extracellular matrix or target cells can extend to reach a larger size with a diameter between 10-15 um.

2.       Line 283:  If T cells are directly activated by NETs, please provide the reference. If T cell activation is subsequent to the activation of pDCs by NETs, it should be removed here. The current presentation sounds like NETs directly activate both pDCs and T cells.

3.       While literatures support antigen presenting capability of neutrophils, antigen presentation is usually not a major function of neutrophils due to their short life span which prevents long-lasting antigen presentation till T cells are recruited to the site. Activation of T cells by neutrophils is usually not mediated through direct antigen presentation but rather by cytokines, chemokines and other molecules.

4.       Line 328-329: need reference for “Arginine is important for maintaining the activities of T-cells through expression of T-cell co-receptor CD3ζ.”

5.       Line 330: CAT1 and 2 are transporters for arginine not arginase I.

6.       Line 331-332: I am not sure what’s attempted to be pointed out here. Arginine consumption by T cells enhances anti-tumor capacity of T cells as arginine and tryptophan are the two key amino acids for T cell activation. See comment 4 above. Elevation of nitric oxide in the tumor microenvironment is multifactorial and metabolism of arginine through NOS in T cells is not the only factor. In addition, nitric oxide has biphasic roles in tumor immunity meaning high and low NO concentration has opposite effects on tumor immune responses. Therefore, this needs some clarifications.

7.       Line 616-620: Need clarifications on “TLR ligands can activate immune cells by binding to TLRs, which are expressed on the surface of various immune cells, including dendritic cells and T cells. By loading NDEs with TLR ligands, it is possible to stimulate the immune response and enhance anti-tumor immunity. NDEs loaded with a TLR9 ligand could activate dendritic cells and induce anti-tumor immunity in a mouse model of melanoma.”

There are TLRs expressed on the surface of cells and endosomal TLRs that are expressed intracellularly within the endosomes. The authors describe TLRs as surface receptors but then use intracellular TLR9 as an example to demonstrate the importance of TLRs. This creates confusions.

8.       Figure 1 needs some more labelling or modification. During promyelocyte stage, azurophilic granules are the dominant granules but these will be diluted during later developmental stages when other specific, secretory and gelatinase granules are formed. If the authors wish to present the granules in such details, it should be correctly drawn. If not, one can just draw all granules in one single color throughout the granulopoiesis.

9.       In Fig.2, is iNOS also secreted by neutrophils into exosomes?

Author Response

Responses to reviewer

This review article by Abhishek Dutta et. Al. is well presented but some concerns are addressed below.

  1. Line 231: The size of neutrophils needs to be clearly defined. Neutrophils once entering tissues and attached to extracellular matrix or target cells can extend to reach a larger size with a diameter between 10-15 um.

First, we would like to thank the reviewer for reading the manuscript thoroughly and pointing out the errors and updating the gaps.

As per your suggestion, we have updated the text. “(~8.85+ 0.44 µM in diameter) in human (https://doi.org/10.3390/ijms21124523), however, once they enter tissues and attach to extracellular matrix or target cells, neutrophils can attain larger size and shape (10µM-15µM) (https://doi.org/10.1038/s12276-019-0227-1)”. An extravasating neutrophil becomes elongated due to microparticle deposition. LFA-1 (an integrin, CD11a/CD18) is localized at the trailing edge of neutrophils during the elongation step of extravasation upon stimulation with fMLP and the cell length repeatedly become longer until the trailing edge is finally retracted the endothelial basement membrane (https://doi.org/10.1038/s12276-019-0227-1). The average range of cancer cell size varies from 15-25 µm in diameter (https://doi.org/10.1016/j.addr.2018.01.002) which is comparatively larger than neutrophils and hence they engulf the cancer cells via trogocytosis. We have also incorporated the necessary changes in the manuscript based on current reports in Line 231.

  1. Line 283: If T cells are directly activated by NETs, please provide the reference. If T cell activation is subsequent to the activation of pDCs by NETs, it should be removed here. The current presentation sounds like NETs directly activate both pDCs and T cells.

NETs released by human neutrophils can directly prime T cells by reducing their activation threshold with up-regulation of CD25 and CD69 on T cells, and NETs/cell contact and TCR signaling is needed (https://doi.org/10.1186/s12964-019-0471-y) (https://doi.org/10.1038/s41467-022-28172-4). We have included the reference “https://doi.org/10.4049/jimmunol.1103414”. In this article authors have shown that T lymphocytes can be directly primed through NETs.

  1. While literatures support antigen presenting capability of neutrophils, antigen presentation is usually not a major function of neutrophils due to their short life span which prevents long-lasting antigen presentation till T cells are recruited to the site. Activation of T cells by neutrophils is usually not mediated through direct antigen presentation but rather by cytokines, chemokines and other molecules.

We completely agree with the reviewer on this aspect of neutrophil viability and short lifespan. However recently, we encountered an article suggesting that neutrophils were long thought to be short-lived cells of the innate immune system; however, there is increasing evidence that they can survive for several days and that under several conditions they can acquire classical APC markers and functions (https://doi.org/10.1189/jlb.1MR1014-502R) (https://doi.org/10.1046/j.1365-2249.2003.02245.x). Human neutrophils present the antigen to antigen-specific memory CD4+T cells in an HLA-DR–dependent manner. In the mentioned study, they investigated whether neutrophils are viable for a sufficient time and can acquire antigen presentation capacity to contribute to adaptive immunity. They found that neutrophils are able to present cognate antigens to autologous memory CD4+ T cells in vitro and ex vivo although to a lower extent compared with MDCs and monocytes (https://doi.org/10.1182/blood-2016-10-744441). A recent report demonstrated that resting neutrophils suppress T cell proliferation, activation and cytokine production but that de-granulating neutrophils do not, and neutrophil-released intracellular contents enhance proliferation (https://doi.org/10.3389/fimmu.2021.633486). Also, T cells early in the activation process are susceptible to suppression by neutrophils, while later-stage T cells are not, and naïve T cells do not respond at all.

We agree with your comments. Neutrophils are not professional APCs. Cytokine or chemokine functions are the primary mechanisms by which T cells are activated by neutrophils. Occasionally, such as autoimmune disorders, inflammation, phagocytosis of IgG-opsonization erythrocytes (https://doi.org/10.1182/bloodadvances.2018028753) neutrophils acquire antigen presentation characteristics, which can prime T cells. According to a study published in 2021 (https://doi.org/10.3389/fimmu.2021.633486) naive T cells don't respond to neutrophils, while CD3/CD28 stimulated T cells (early stimulated) are suppressed by neutrophils. Neutrophils activate late stimulated T cells (stimulated for 24 hours by CD3/CD28). Another study (https://doi.org/10.1182/blood-2006-12-063826) by C. Beauvillain demonstrated cross-presentation of ovalbumin by neutrophils to CD8+ T cells through MHC Class I in OT1 transgenic mice.

  1. Line 328-329: need reference for “Arginine is important for maintaining the activities of T-cells through expression of T-cell co-receptor CD3ζ.”

We have included the reference.

  1. Line 330: CAT1 and 2 are transporters for arginine not arginase I.

Thank you for the correction. We apologize, it was a typo error.

  1. Line 331-332: I am not sure what’s attempted to be pointed out here. Arginine consumption by T cells enhances anti-tumor capacity of T cells as arginine and tryptophan are the two key amino acids for T cell activation. See comment 4 above. Elevation of nitric oxide in the tumor microenvironment is multifactorial and metabolism of arginine through NOS in T cells is not the only factor. In addition, nitric oxide has biphasic roles in tumor immunity meaning high and low NO concentration has opposite effects on tumor immune responses. Therefore, this needs some clarifications.

We thank the reviewer for raising the bidirectional role of NO, which is in fact supportive of neutrophil plasticity. We have already included the immunosuppressive effect of Arginine consumption by conversion of CD4+CD25- cells into CD4+CD25+ cells regulatory T cells through metabolic conversion of arginine to NO. Furthermore, on your due approval, we are adding this comment in the same part for clarification with references.

In this section we wanted to highlight the role of Arg1 released from neutrophils which depleted arginine from the microenvironment. This reduced concentration of arginine leads to T cells suppression. The concentration of L-arg in the core regions of solid tumors is about 5 times lower as compared with tumor periphery and this difference turned out to be the highest among all of the measured amino acids. Liu et al. showed that myeloid cells in the tumor microenvironment express arginase and suppress cytotoxic T lymphocyte (CTL) activity in NO-independent manner (https://doi.org/10.3389/fimmu.2020.00938). Arginine can also be metabolized by NOS to produce (nitric oxide) NO. NO can induce CD4+CD25+ Foxp3− regulatory T cells from CD4+CD25− T cells via p53, IL-2, and OX40 (https://doi.org/10.1073/pnas.0703725104). NO can also suppress T cells via Stat5 inhibition (https://doi.org/10.1182/blood-2006-02-002246). Low exposure to NO may promote tumor progression and metastasis by direct induction of tumor cells proliferation, migration, and invasion and indirectly through the expression of angiogenic factors in tumor cells (https://doi.org/10.1038/sj.cr.7290133). Although, high concentrations of NO derived from iNOS in macrophages induce p53 phosphorylation resulting in endothelial cell growth arrest, and higher concentrations and prolonged exposure time induce cell death. Prolonged production of NO has been associated with the release of cytochrome c from the mitochondria, activation of caspase, modulation of anti-apoptotic Bcl-2 proteins, and increase in p53 expression. A recent study has reported that one of the underlying mechanisms by which NO-mediated NF-κB inhibition suppresses tumor cell resistance and metastasis is through inhibition of the downstream targets Snail and the transcription factor Yin Yang 1 (YY1) and induction of Raf-1 kinase inhibitor protein (RKIP). Therefore, the function of NO within the tumor microenvironment is much debated therefore needs further research. Overall, we can safely say that NO is a 'Doubled-Edged Sword' in cancer.

In this review we are focusing on the role of neutrophils derived Arginase 1 and how it depletes arginine from the micro-environment to suppress the immune cells. To emphasize the process of arginine metabolism and how its downstream product can also suppress immune cells we considered the role of NO in immune suppression.

  1. Line 616-620: Need clarifications on “TLR ligands can activate immune cells by binding to TLRs, which are expressed on the surface of various immune cells, including dendritic cells and T cells. By loading NDEs with TLR ligands, it is possible to stimulate the immune response and enhance anti-tumor immunity. NDEs loaded with a TLR9 ligand could activate dendritic cells and induce anti-tumor immunity in a mouse model of melanoma.”

There are TLRs expressed on the surface of cells and endosomal TLRs that are expressed intracellularly within the endosomes. The authors describe TLRs as surface receptors but then use intracellular TLR9 as an example to demonstrate the importance of TLRs. This creates confusion.

Thank you for your comment. We agree the statement needs further clarification. As suggested by the Reviewer we have now modified the statement to make it more legible. We have also added how exosomes loaded with ligands for surface TLRs can stimulate the immune cells and we have also added relevant references for the same. TLR9 is mainly present intracellularly within endosomes. According to a 2004 study, 2-10 % of PBMCs express TLR9 on cell surfaces (https://doi.org/10.1128/IAI.72.12.7202-7211.2004). Colonic epithelial cells were also reported to express TLR-9 on the cell surface post exposure to DNA from various commensal and pathogenic microbes (https://doi.org/10.1128/IAI.01662-06). There are many reports claiming endosomal TLRs can also be present on the cell surface (https://doi.org/10.3389/fimmu.2020.620972). The response of the host may be enhanced by endosomal TLRs on the cell surface.

  1. Figure 1 needs some more labelling or modification. During promyelocyte stage, azurophilic granules are the dominant granules but these will be diluted during later developmental stages when other specific, secretory and gelatinase granules are formed. If the authors wish to present the granules in such details, it should be correctly drawn. If not, one can just draw all granules in one single color throughout the granulopoiesis.

As per your suggestion, the granules in the figure have been modified to a uniform color.

  1. In Fig.2, is iNOS also secreted by neutrophils into exosomes?

Thank you for your comment. Figure 2 illustrates the role of neutrophils of subtype N1 and N2. It is not related to neutrophils' exosomes. However, circulating exosomes from septic shock patient have been shown to contain NOS (https://doi.org/10.1186/cc6176). However, the source was platelets. No report is available identifying iNOS in NDEs.

Round 2

Reviewer 2 Report

Thanks for the revision.